# Integral assays of hemostasis in hospitalized patients with COVID-19 on admission and during heparin thromboprophylaxis

Andrey Y. Bulanov[1], Ekaterina L. Bulanova[1], Irina B. Simarova[1], Elizaveta A. Bovt[2,3]*, Olesya O. Eliseeva[4], Soslan S. Shakhidzhanov[2,3], Mikhail A. Panteleev[2,3,4,5], Aleksandr G. Roumiantsev[2], Fazoil I. Ataullakhanov[2,3,4,5], Sergey S. Karamzin[3]

1 Moscow City Clinical Hospital 52 of Healthcare Department, Moscow, Russia, 2 Dmitry Rogachev National Medical Research Center Of Pediatric Hematology, Oncology and Immunology, Moscow, Russia, 3 Center for Theoretical Problems of Physicochemical Pharmacology, Moscow, Russia, 4 Faculty of Biological and Medical Physics, Moscow Institute of Physics and Technology, Dolgoprudnyi, Russia, 5 Faculty of Physics, Lomonosov Moscow State University, Moscow, Russia

* ie.bovt.rv@gmail.com

## Abstract

### Background

Blood coagulation abnormalities play a major role in COVID-19 pathophysiology. However, the specific details of hypercoagulation and anticoagulation treatment require investigation. The aim of this study was to investigate the status of the coagulation system by means of integral and local clotting assays in COVID-19 patients on admission to the hospital and in hospitalized COVID-19 patients receiving heparin thromboprophylaxis.

### Methods

Thrombodynamics (TD), thromboelastography (TEG), and standard clotting assays were performed in 153 COVID-19 patients observed in a hospital setting. All patients receiving treatment, except extracorporeal membrane oxygenation (ECMO) patients (n = 108), were administered therapeutic doses of low molecular weight heparin (LMWH) depending on body weight. The ECMO patients (n = 15) were administered unfractionated heparin (UFH).

### Results

On admission, the patients (n = 30) had extreme hypercoagulation by all integral assays: TD showed hypercoagulation in ~75% of patients, while TEG showed hypercoagulation in ~50% of patients. The patients receiving treatment showed a significant heparin response based on TD; 77% of measurements were in the hypocoagulation range, 15% were normal, and 8% remained in hypercoagulation. TEG showed less of a response to heparin: 24% of measurements were in the hypocoagulation range, 59% were normal and 17% remained in hypercoagulation. While hypocoagulation is likely due to heparin treatment, remaining in significant hypercoagulation may indicate insufficient anticoagulation for some patients, which is in agreement with our clinical findings. There were 3 study patients with registered

**Data Availability Statement:** All relevant data are within the paper and its Supporting Information files.

**Funding:** This research was funded by the Russian Science Foundation (21-45-00012) and by the Russian Foundation for Fundamental Investigations (19-34-90036).

**Competing interests:** The authors have declared that no competing interests exist.

thrombosis episodes, and all were outside the target range for TD parameters typical for effective thromboprophylaxis (1 patient was in weak hypocoagulation, atypical for the LMWH dose used, and 2 patients remained in the hypercoagulation range despite therapeutic LMWH doses).

## Conclusion

Patients with COVID-19 have severe hypercoagulation, which persists in some patients receiving anticoagulation treatment, while significant hypocoagulation is observed in others. The data suggest critical issues of hemostasis balance in these patients and indicate the potential importance of integral assays in its control.

## Introduction

Blood coagulation abnormalities are known to be associated with and play a significant role in the pathogenesis of viral pneumonia. This recently became a subject of major concern with regard to the COVID-19 pandemic; early reports with Chinese patients indicated an association between increased D-dimer and prolonged clotting time assays and mortality [1, 2].

It has been shown that COVID-19 infection results in high incidence of thromboembolic complications especially among severely ill patients [3, 4]. Autopsy studies of COVID-19 patients have shown that microthrombi may be found in different organs and pulmonary macro and microthrombi are the most frequent of them [5, 6]. Anticoagulation treatment was shown to improve survival in severely ill patients [7], along with reduction in thromboembolic complication rates [8–10]. Several factors which can lead to hypercoagulability have been found in plasma of COVID-19 patients [11]: e.g. elevated tissue-factor levels, anti-phospholipid antibodies, and protein C system dysregulation. However, those factors can differ between patients and their contribution may change during treatment.

The overall procoagulant pattern of increased D-dimer was confirmed for COVID-19 patients worldwide [12]. However, low specificity of D-dimer makes it difficult to use this assay as a reliable indicator of hypercoagulability and thromboembolic event presence because its elevation could be the result of active fibrinolysis of the thrombi which had already been formed. It can also be attributed to its nonspecific elevation in diabetes, some liver and kidney diseases, or because of fibrinolysis outside the vessel lumen [13–16]. Other plasma coagulation assays such as aPTT and PT were normal or hardly prolonged [17–20] and no reliable results were reported on their correlation with illness severity or thromboembolism presence. While fibrinogen levels are also generally elevated in patients, no substantial evidence of its role in plasma hypercoagulability were found yet [21].

The only class of coagulation assays attempting to directly determine a state of coagulability are so-called global assays of hemostasis, such as thromboelastography, thrombin generation, and thrombodynamics [22]; these assays aim to imitate the major aspects of in vivo clotting and detect shifts in the overall balance, including subtle hypercoagulative changes [13, 23–25] and therapy monitoring [25–27].

TG, TEG and ROTEM assays detect patient plasma hypercoagulability [28, 29], but there is no clearly established relationship between their values and thrombotic complications during the disease [30–32], and no data were reported on anticoagulation treatment influence on the results of these tests along with changes in coagulability it induces.

In this work, we used two different integral assays to investigate the status of the coagulation system in COVID-19 patients receiving heparin.

## Methods

### Patients

Individuals admitted consequently to Moscow City Clinical Hospital 52 between April 2020 and June 2020 with PCR-confirmed COVID-19 infection were enrolled in the study. For the purposes of the research the data was collected prospectively and were fully anonymized prior to analysis. Patients were over the age of 18, and the study was approved by the Independent Ethical Committee of Dmitriy Rogachev National Medical Research Center of Pediatric Hematology, Oncology, and Immunology, Ministry of Healthcare of Russia, Moscow. Informed written or verbal consents in the case if a patient was unable to give written consent for taking part in the study, along with usage of medical records and samples were given by all patients. Consent for additional sample collection for thrombodynamics assay was obtained from all participants of the study.

According to the recommendations of the Ministry of Health anticoagulation therapy was prescribed to all patients with COVID-19 if no contraindications such as low platelet counts, bleeding, or florid renal failure were present. All patients receiving treatment were administered medium-dose LMWH depending on body weight according to the routine hospital protocol for COVID-19 patients; enoxaparin (100 ME/kg twice daily), dalteparin (100 ME/kg twice daily), or fondaparinux (5 mg– 10 mg/day depending on body weight). Patients receiving ECMO were administered unfractionated heparin (250–1300 units/hour depending on weight and bleeding tendency).

On admission to the hospital blood for TD and TEG assays was collected before any anticoagulation was given to a patient. Further coagulation tests were mostly performed at the peak of heparin treatment (median time of blood sampling after LMWH injection was 3.5 h, interquartile range—0.5 h). For the TD and TEG assays, only measurements performed at the peak of heparin treatment were included in the group comparison analysis. For routine assays (APTT, PT, INR, Fg), all data points were used for the group comparison analysis and correlation analysis.

### Patient state severity scaling

Based on the National Early Warning Score (NEWS) [33], all patients were assigned to one of three groups based on the severity of their state: moderate (1–4 points), severe (5–6 points), or critical (>7 points).

### Evaluation of respiratory failure

All patients were divided into the following three groups:

PaO2 60–79 mmHg; SaO2 90–94%; without mechanical ventilation and without ECMO.

PaO2 40–59 mm Hg; SaO2 75–89%; with mechanical ventilation and without ECMO.

PaO2 below 40 mmHg; SaO2 below 75%; with mechanical ventilation and ECMO.

### Blood collection and processing

For patients in the intensive care unit, blood was taken from a central venous catheter washed with saline solution. Patients in the admission, nephrology and cardiology departments had blood collected in 4.3 ml vacuum tubes (S-monovette, Sarstedt, Germany) with 3.2% sodium

citrate. Platelet-poor plasma was obtained by centrifugation for 15 min at 1600 g, and part of the plasma was repeatedly processed by centrifugation at 10,000 ×g for 5 min to obtain platelet-free plasma for use in the thrombodynamics assays [34].

## Thrombodynamics

The thrombodynamics assay [35, 36] was performed using a Thrombodynamics Analyzer System T2-F and Thrombodynamics TDX kit (HemaCore LLC, Moscow, Russia). The coagulation process in thrombodynamics starts from a localized surface that has immobilized tissue factor mimicking blood vessel wall damage. The process of fibrin clot formation was recorded in time-lapse video microscopy mode by means of the dark-field light scattering method. The obtained series of photos showed how the form, size, and density of the fibrin clot changed over time. On the basis of the recorded photos, the numerical parameters of the spatiotemporal dynamics of fibrin clot formation were calculated as follows: Vi (the initial rate of clot growth calculated as the mean clot growth rate of the interval 2–6 minutes after the beginning of clot growth); Vs (the stationary rate of clot growth calculated as the mean clot growth rate of the interval 15–25 minutes after the beginning of clot growth); CS (clot size at the 30th minute of measurement); D (the density of the formed clot–an optical parameter equal to the intensity of light scattering from a fibrin clot, proportional to the density of the fibrin mesh); and Tsp (time of spontaneous clot formation) [26].

## Thromboelastography

The samples for most patients were evaluated by TEG using nonkaolin-activated citrated blood. TEG with kaolin-activated citrated blood was performed for ECMO patients (according to the routine hospital protocol for ECMO patients). This study utilized a TEG 5000 Thrombelastograph Hemostasis Analyzer system and disposable cups (Haemonetics Corporation, MA, USA). Recalcification was carried out with 20 μl of 0.2 M $CaCl_2$. Four TEG variables (reaction time [R], clot formation time [K], clot development kinetics [Angle], and maximum amplitude [MA]) were analyzed for all assays. The reaction time, R, denotes the latency from mixing until the clot starts to form (2-mm amplitude), the clot formation time (k) represents the time taken by the forming clot to reach a fixed degree of viscoelasticity, the Angle parameter (a) represents the velocity of clot formation, and MA represents the maximum strength of the clot.

## Other assays

Other routinely available in the hospital setting coagulation assays APTT, PT, INR, and fibrinogen were performed with a Sysmex CS 2100i automated coagulometer (Sysmex Corporation, Kobe, Japan), ACL TOP 500 (Instrumentation Laboratory, MA, USA) and ACL TOP 300 (Instrumentation Laboratory, MA, USA).

## Statistics

All data obtained during the study was used, and no outliers were removed before the analysis. For groups comparison non-parametric Mann-Whitney U-criterion was used at the significance level of 0.01. For correlation analysis Spearman's correlation coefficient was calculated at the significance level of 0.01. Box plots display a median, a first and a third quartiles, and the length of the whiskers corresponds to the 1.5 interquartile range. All the calculations were performed with the raw table data by self-written C# program and then double checked by means of SciPy–free and open-source Python library used for scientific computing and technical computing (https://www.scipy.org).

# Results

## Patient demographics

The total number of patients enrolled in the study was 153, and their characteristics are presented in Table 1. The two main groups were those tested upon arrival, prior to heparin therapy (30 patients), and those tested while in the hospital (123 patients), who all received heparin therapy. All patients belonged to the middle-age or elderly age category (48–74 years). Among the 123 patients in the hospital, there were two possible subgroups, those receiving ECMO (15 patients) and those without ECMO (108 patients), along with three different degrees of patient severity and three different degrees of respiratory failure. In 13 of these patients, nonfatal venous thromboembolism events (including 2 pulmonary embolisms) were recorded during treatment, and in 9 patients, nonfatal bleeding of various severities (including 3 gastrointestinal and 2 pulmonary events) was recorded.

Among the patients, there were 3 thromboses in the non-ECMO treatment group: one deep vein thrombosis, one pulmonary embolism, and one posterior tibial vein thrombosis, which occurred at 8, 4, and 6 days following the analysis, respectively. The first two patients were discharged, while the third continued to receive treatment at the end of the study. There were 3 bleeding events, all of which occurred among the ECMO patients. Two of them had intrapleural bleeding, and the third had nose bleeding. In all 3 patients, bleeding occurred 1 day after analysis, and the patients had fatal outcomes 1, 3, and 7 days after the analysis.

## Thrombodynamics results

Parameters of thrombodynamics in the assayed groups are summarized in Fig 1. On admission, the patients had extreme hypercoagulation, with the main parameters exceeding the upper limit of the normal range in the vast majority of patients. The most profound effects were in regard to clot size (75% of patients in hypercoagulation) and initial (in 83%) and stationary (in 76%) clot growth rates (Table 2).

The patients receiving treatment, as a whole group as well as the subgroups, had wide distributions among the thrombodynamics parameters that exceeded both the lower and upper limits of the reference ranges. Patients receiving treatment generally showed a significant heparin response based on Vs (the parameter most sensitive to heparin); 77% of TD measurements were in the hypo range, 15% were normal and 8% remained in hypercoagulation (Fig 1D, first plot). The presence of both hypo- and hypercoagulation indicates the efficiency of heparin treatment and, at the same time, suggests that at least some patients (up to 23% based on Vs; Fig 1D, first plot) did not have their anticoagulation suppressed. The parameters of ECMO patients were further shifted toward hypocoagulation compared with those without ECMO, likely due to the effect of unfractionated heparin treatment.

Similarly, increased severity of the disease (Fig 1C–1F, second column of plots) and the degree of respiratory failure (Fig 1C–1F, third column of plots) were associated with hypocoagulation. The possible reasons for stronger hypocoagulation in these groups were increased doses of heparin used in severe and critical patients and lower platelet counts in critical patients (mostly because of ECMO).

## Thromboelastography results

The citrated native TEG parameters (Fig 2A) of patients on admission also showed significant hypercoagulation; R, K, Angle and MA were beyond the normal range in 24%, 17%, 48%, and 54% of the patients, respectively. In patients receiving treatment, the TEG parameters sensitive to heparin showed less of a response than the stationary clot growth rate in TD; the Angle

**Table 1. Descriptive statistics.**

| Parameter (normal range) | Before treatment | On treatment | ECMO | |
|---|---|---|---|---|
| | | | **No** | **Yes** |
| Number of patients | 30 | 123 | 108 | 15 |
| Age | 68 (59–74) | 58 (48–68) | 60 (51–69) | 57 (47–59) |
| Heparin daily dosage, IU | 0 | 16000 (10000–20200) | 15000 (10000–16000) | 18000 (10500–24000) |
| **Thrombodynamics** (for patients on treatment—measured at the peak of the heparin effect) | | | | |
| Measurements total | 30 | 180 | 89 | 91 |
| Vi, um/min (38–56) | 59.8 (58–64.4) | 45.8 (31.2–54.7) | 50.6 (42.1–58.2) | 36 (21.4–49.5) |
| Vs, um/min (20–29) | 32.6 (29.6–35.8) | 12 (9.1–18.8) | 14.1 (9.4–20.1) | 11 (8.2–15.5) |
| CS, um (800–1200) | 1322 (1196–1420) | 729 (553–977) | 845 (684–1035) | 619 (454–839) |
| D, a.u. (15000–32000) | 30003 (26336–32504) | 27484 (23839–31922) | 29526 (26834–33418) | 22799 (21110–26505) |
| **TEG: citrated native** (for patients on treatment—measured at the peak of the heparin effect) | | | | |
| Measurements total | 29 | 84 | 84 | 0* |
| R, min (9–27) | 10.7 (9.1–12.7) | 14.5 (10.8–22.7) | 14.5 (10.8–22.7) | - |
| K, min (2–9) | 2.5 (2.2–2.9) | 4.9 (2.9–8.8) | 4.9 (2.9–8.8) | - |
| Angle, deg (22–58) | 56.3 (49.9–59.9) | 39.4 (22.8–55.8) | 39.4 (22.8–55.8) | - |
| MA, mm (44–64) | 65.3 (61.2–68.7) | 65.2 (55.2–73.9) | 65.2 (55.2–73.9) | - |
| **TEG: citrated kaolin** (for patients on treatment—measured at the peak of the heparin effect) | | | | |
| Measurements total | 0* | 91 | 0* | 91 |
| R, min (2–8) | - | 10.3 (6.9–15.6) | - | 10.3 (6.9–15.6) |
| K, min (1–3) | - | 2.7 (2–4.9) | - | 2.7 (2–4.9) |
| Angle, deg (55–78) | - | 52.9 (34–63.2) | - | 52.9 (34–63.2) |
| MA, mm (51–69) | - | 60.9 (51.4–66.2) | - | 60.9 (51.4–66.2) |
| **Routine coagulation tests** | | | | |
| Measurements total | 30 | 225 | 122 | 103 |
| Fibrinogen, g/l (2,67–4,37) | 5.85 (5.27–6.88) | 5.72 (4.3–7.05) | 6.3 (4.63–7.22) | 5.01 (3.82–6.43) |
| Prothrombin time, sec (11,5–14,8) | 15.9 (15–17.5) | 17.1 (15.1–18.5) | 15.5 (13.4–17.2) | 18 (17–19.6) |
| INR (0,9–1,2) | 1.24 (1.16–1.38) | 1.34 (1.2–1.46) | 1.25 (1.14–1.38) | 1.42 (1.33–1.56) |
| APTT, sec (25,1–36,5) | 31.6 (28.2–32.6) | 32 (28.5–36.6) | 30.3 (27.1–33) | 35 (31–46.3) |

| Parameter (normal range) | Severity | | | Respiratory failure | | |
|---|---|---|---|---|---|---|
| | **Moderate** | **Severe** | **Critical** | **I** | **II** | **III** |
| Number of patients | n/a | n/a | n/a | n/a | n/a | n/a |
| Age | n/a | n/a | n/a | n/a | n/a | n/a |
| Heparin daily dosage, IU | 15000 (10000–16000) | 16800 (11400–24000) | 18000 (10800–24000) | 15000 (10000–16000) | 16000 (10000–20000) | 18000 (10500–24000) |
| **Thrombodynamics** (for patients on treatment—measured at the peak of the heparin effect) | | | | | | |
| Measurements total | 62 | 89 | 29 | 47 | 42 | 91 |
| Vi, um/min (38–56) | 50.5 (42.2–58.1) | 42.9 (30.2–54.7) | 25.2 (19.1–40.3) | 50.6 (41.9–59.2) | 49.8 (42.3–56.2) | 36 (21.4–49.5) |
| Vs, um/min (20–29) | 13 (9.3–20.4) | 13.1 (9.3–20.3) | 8.5 (6.7–12.5) | 14 (9.3–20.1) | 15 (9.8–19.8) | 11 (8.2–15.5) |
| CS, um (800–1200) | 826 (680–1045) | 735 (553–987) | 483 (414–694) | 871 (703–1049) | 825 (680–1024) | 619 (454–839) |
| D, a.u. (15000–32000) | 30029 (27739–33229) | 25821 (22330–31370) | 21844 (20399–24408) | 31076 (27815–33219) | 29016 (25882–33728) | 22799 (21110–26505) |
| **TEG: citrated native** (for patients on treatment—measured at the peak of the heparin effect) | | | | | | |
| Measurements total | 57 | 26 | 1 | 42 | 42 | 0 |
| R, min (9–27) | 13.7 (10.5–24.9) | 15 (13.1–18.7) | 14.3 (14.3–14.3) | 13 (10.3–28.1) | 15 (12.9–20) | - |
| K, min (2–9) | 4.4 (2.8–8.8) | 5.6 (3.2–9.2) | 2.6 (2.6–2.6) | 3.8 (2.8–8.7) | 5.6 (3.1–9.4) | - |

*(Continued)*

**Table 1.** (Continued)

| Parameter (normal range) | Before treatment | On treatment | ECMO | | | |
|---|---|---|---|---|---|---|
| | | | No | Yes | | |
| Angle, deg (22–58) | 42.9 (22.1–57.6) | 33.9 (29.6–50.9) | 63.5 (63.5–63.5) | 45.4 (24–57.9) | 35.3 (22.1–54.5) | - |
| MA, mm (44–64) | 66.7 (58.8–74.6) | 64.6 (52.1–71.3) | 79.5 (79.5–79.5) | 65.7 (58.5–74.1) | 65 (52.8–73.6) | - |
| **TEG: citrated kaolin** (for patients on treatment—measured at the peak of the heparin effect) | | | | | | |
| Measurements total | 0 | 64 | 27 | 0 | 0 | 91 |
| R, min (2–8) | - | 9 (6.8–14.2) | 12.8 (9.9–26.6) | - | - | 10.3 (6.9–15.6) |
| K, min (1–3) | - | 2.4 (1.8–3.7) | 4.2 (2.5–7.2) | - | - | 2.7 (2–4.9) |
| Angle, deg (55–78) | - | 56.1 (40.3–65) | 38.7 (26.1–53.9) | - | - | 52.9 (34–63.2) |
| MA, mm (51–69) | - | 62.3 (54.6–68) | 58.1 (42.3–61.1) | - | - | 60.9 (51.4–66.2) |
| **Routine coagulation tests** | | | | | | |
| Measurements total | 81 | 110 | 34 | 65 | 57 | 103 |
| Fibrinogen, g/l (2,67–4,37) | 6.4 (4.9–7.22) | 5.19 (4.03–6.79) | 4.71 (3.72–5.49) | 6.7 (5–7.25) | 6.1 (4.5–7.2) | 5.01 (3.82–6.43) |
| Prothrombin time, sec (11,5–14,8) | 15.1 (13.4–16.9) | 17.5 (15.8–18.8) | 17.9 (17.1–19.9) | 15.1 (13.5–17.1) | 15.7 (13–17.4) | 18 (17–19.6) |
| INR (0,9–1,2) | 1.24 (1.13–1.36) | 1.38 (1.27–1.49) | 1.41 (1.34–1.58) | 1.23 (1.13–1.37) | 1.28 (1.19–1.39) | 1.42 (1.33–1.56) |
| APTT, sec (25,1–36,5) | 30.6 (27.8–33) | 32.7 (29.1–39.1) | 35.8 (30.7–48.5) | 30.4 (27.2–33.4) | 30 (27.2–32.8) | 35 (31–46.3) |

continuous data: median (interquartile range); other values are presented as the number

n/a—not applicable, as same patient may change group during the treatment.

*TEG with kaolin-activated citrate blood was performed only for patients on ECMO, for all other patients TEG citrated native was performed

parameter of citrated native TEG showed hypocoagulation in 24% of measurements only. Similar to thrombodynamics, a significant portion of patients remained in the hypercoagulation range in TEG (17% based on the Angle parameter and 60% based on the MA parameter). TEG citrated kaolin was performed only for the ECMO patients (Fig 2B). R and Angle for the TEG citrated kaolin parameters showed hypocoagulation in 63% and 54% of patients, respectively. Increased severity of the disease (Fig 2A and 2B, second column of plots) and the degree of respiratory failure (Fig 2A and 2B, third column of plots) were associated with hypocoagulation in the TEG assays.

## Clotting assay results

In contrast to the integral assays, clotting time assays (Fig 3) in patients on admission were either normal or showed hypocoagulation. Only fibrinogen was increased (Fig 3A, first plot). The distribution did not shift greatly for patients receiving treatment without ECMO (Fig 3, first column of plots). For those receiving ECMO, there were shifts in APTT, PT, and INR toward hypocoagulation, and fibrinogen was decreased (Fig 3, first column of plots).

It is noteworthy that in most patients, both at admission and during treatment, the level of fibrinogen was significantly increased (87% and 80%, respectively). At the same time, the parameters of global tests reflecting the level and functional activity of fibrinogen were increased in a much smaller number of patients. Thus, the parameter D clot density in the TD test was increased in only 33% of new patients and 25% of patients on therapy. The Angle parameter of citrated native TEG for patients on therapy was increased in only 17% of cases.

## Correlation between integral assays

A significant correlation of approximately 0.7 was observed for the main thrombodynamics parameters (Vi, Vs, CS) and the TEG Angle parameter (Table 3, Fig 4A and 4B). Their

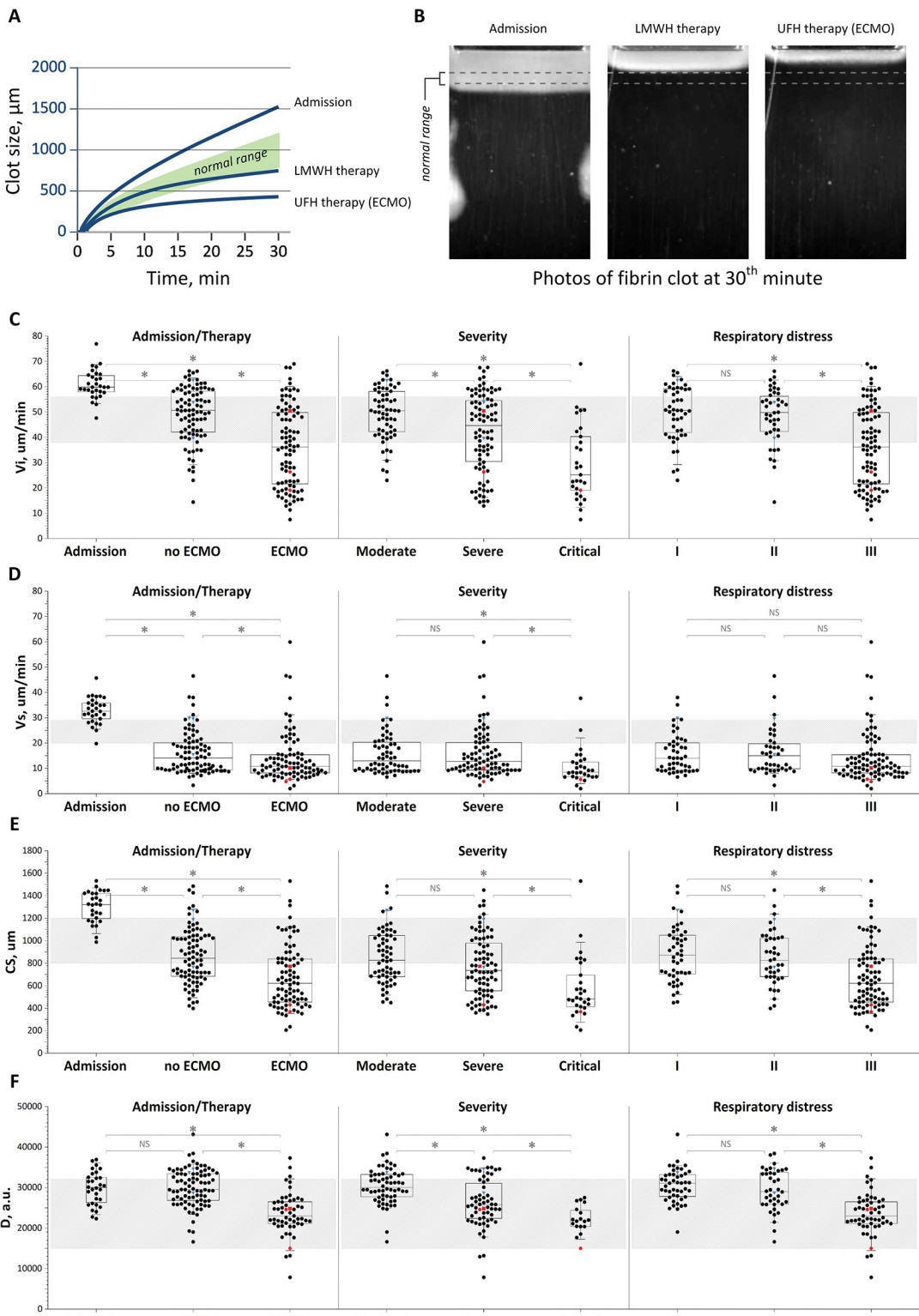

**Fig 1. Thrombodynamics assay parameters in the patient groups.** (A) Examples of clot size as a function of time curves for patients on admission and during treatment and (B) images of clots for representative experiments. (C) Initial clot growth rate Vi, (D) stationary clot growth rate Vs, (E) clot size CS, and (F) clot density D in the patient groups: 1) on admission, with and without extracorporeal membrane oxygenation (ECMO) treatment; 2) depending on patient severity; and 3) depending on the severity of respiratory failure. The patterned area shows the normal reference ranges. The box plot indicates the following

parameters: the median (the horizontal line inside the box), the 25th and 75th percentiles (the bottom and top of the box, respectively), and the 5th and 95th percentiles (the ends of the whiskers); * statistically significant difference (Mann–Whitney U-criterion, p<0.01). NS represents no significant difference. Red circle symbol–bleeding episodes followed in patients receiving heparin therapy, red square symbol–bleeding episode followed in a patient with temporarily suspended heparin therapy, blue triangle symbol–thrombotic episodes followed in patients receiving heparin therapy.

correlation with MA was moderate, and mostly for the citrated native experimental design of TEG. Vi, Vs, and CS were also significantly anticorrelated with K and R at 0.6–0.7. The fibrinogen level greatly (and expectedly) affected clot density D and moderately affected MA (Table 4, Fig 4C–4E) but did not affect other parameters of thrombodynamics or TEG (Table 4). Surprisingly, no correlation was observed between fibrinogen and the TEG Angle parameter, which is supposed to be the most sensitive to fibrinogen levels.

## Integral assays and clinical outcomes

In most patients, after starting heparin therapy, global hemostasis tests record the target state of hypocoagulation, up to 76% according to TD, as the most sensitive global test for heparin effects [26, 35]. However, two risk groups are also clearly distinguished: 1) patients with a risk of thrombotic complications for whom hypocoagulation has not been achieved according to global tests (14.7% of TD measurements are normal, and 8.5% are in the hypercoagulation zone); and 2) patients at risk of bleeding for whom a standard dose of heparin has led to an excessive suppression of coagulation.

**Table 2. Patients distribution according to coagulation status on admission and on therapy.**

| Parameter | On admission | | | On therapy | | |
|---|---|---|---|---|---|---|
| | Hypo, % | Norma, % | Hyper, % | Hypo, % | Norma, % | Hyper, % |
| Thrombodynamics | | | | | | |
| Vi (um/min) | 0,0 | 16,7 | 83,3 | 33,9 | 43,3 | 22,8 |
| Vs (um/min) | 3,5 | 20,7 | 75,9 | 76,8 | 14,7 | 8,5 |
| CS (um) | 0,0 | 25,0 | 75,0 | 56,1 | 37,0 | 6,9 |
| D (a.u.) | 0,0 | 66,7 | 33,3 | 2,8 | 72,3 | 24,8 |
| TEG: citrated native | | | | | | |
| R (CN) (min) | 0,0 | 75,9 | 24,1 | 19,1 | 72,6 | 8,3 |
| K (CN) (min) | 0,0 | 82,8 | 17,2 | 24,7 | 72,7 | 2,6 |
| Angle (CN) (deg) | 0,0 | 51,7 | 48,3 | 24,4 | 59,0 | 16,7 |
| MA (CN) (mm) | 0,0 | 46,4 | 53,6 | 9,0 | 30,8 | 60,3 |
| TEG: citrated kaolin | | | | | | |
| R (CK) (min) | -* | -* | -* | 62,6 | 37,4 | 0,0 |
| K (CK) (min) | -* | -* | -* | 41,7 | 57,1 | 1,2 |
| Angle (CK) (deg) | -* | -* | -* | 54,6 | 45,5 | 0,0 |
| MA (CK) (mm) | -* | -* | -* | 23,0 | 64,4 | 12,6 |
| Routine tests | | | | | | |
| Fibrinogen (g/l) | 0,0 | 13,3 | 86,7 | 4,3 | 15,7 | 80,0 |
| PT (sec) | 33,3 | 66,7 | 0,0 | 50,6 | 46,0 | 3,5 |
| INR | 43,3 | 56,7 | 0,0 | 58,3 | 41,8 | 0,0 |
| APTT (sec) | 10,0 | 86,7 | 3,3 | 30,2 | 65,3 | 4,5 |

*TEG with kaolin-activated citrate blood was performed only for patients on ECMO

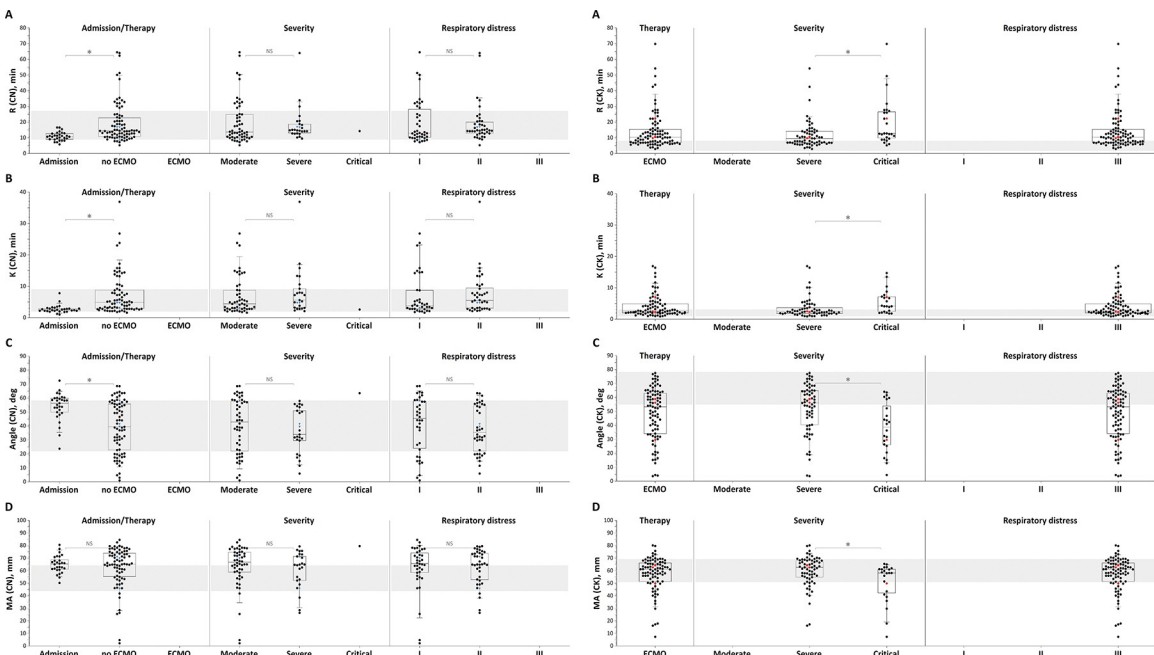

**Fig 2.** a. Thromboelastography (citrated native) parameters in the patient groups. (A) R, (B) K, (C) Angle, and (D) MA in the patient groups: 1) on admission, with and without extracorporeal membrane oxygenation (ECMO) treatment; 2) depending on patient severity; and 3) depending on the severity of respiratory failure. Note that TEG for patients on admission and for those without ECMO was performed with the citrated native version, while TEG for those with ECMO was performed with the native kaolin version. The patterned area shows the normal reference ranges. The box plot indicates the following parameters: the median (the horizontal line inside the box), the 25th and 75th percentiles (the bottom and top of the box, respectively), and the 5th and 95th percentiles (the ends of the whiskers); * statistically significant difference (Mann–Whitney U-criterion, p<0.01). NS represents no significant difference. Red circle symbol–bleeding episodes followed in patients receiving heparin therapy, red square symbol–bleeding episode followed in a patient with temporarily suspended heparin therapy, blue triangle symbol–thrombotic episodes followed in patients receiving heparin therapy. b. Thromboelastography (citrated kaolin) parameters in the patient groups. (A) R, (B) K, (C) Angle, and (D) MA in the patient groups: 1) on admission, with and without extracorporeal membrane oxygenation (ECMO) treatment; 2) depending on patient severity; and 3) depending on the severity of respiratory failure. Note that TEG for patients on admission and for those without ECMO was performed with the citrated native version, while TEG for those with ECMO was performed with the native kaolin version. The patterned area shows the normal reference ranges. The box plot indicates the following parameters: the median (the horizontal line inside the box), the 25th and 75th percentiles (the bottom and top of the box, respectively), and the 5th and 95th percentiles (the ends of the whiskers); * statistically significant difference (Mann–Whitney U-criterion, p<0.01). NS represents no significant difference. Red circle symbol–bleeding episodes followed in patients receiving heparin therapy, red square symbol–bleeding episode followed in a patient with temporarily suspended heparin therapy, blue triangle symbol–thrombotic episodes followed in patients receiving heparin therapy.

For the Vs parameter in the TD test, the target range of effective hypocoagulation was 7–14 μm/min, which was obtained as a result of previous clinical studies and approved by the Ministry of Health of the Russian Federation for practical use. If the Vs parameter is within this range, provided that the blood for analysis is taken at the peak of the drug, therapy is considered effective and safe. With Vs <7 μm/min, the risk of bleeding increases, and with Vs> 14 μm/min, the risk of thrombosis increases.

To assess the predictive capabilities of TDs, the measurement results obtained on the eve of thrombohemorrhagic episodes are of interest. Due to the design features of this observational study, such results were available only for three episodes of thrombosis and three episodes of bleeding (see description in the Patient demographics section above). All of the above measurement points are highlighted in Figures against the background of the points of the remaining measurements. It can be seen in Fig 1D that 2 episodes of bleeding occurred at clot growth rates (Vs) located in the unsafe zone of hypocoagulation (red circle symbol), and the third

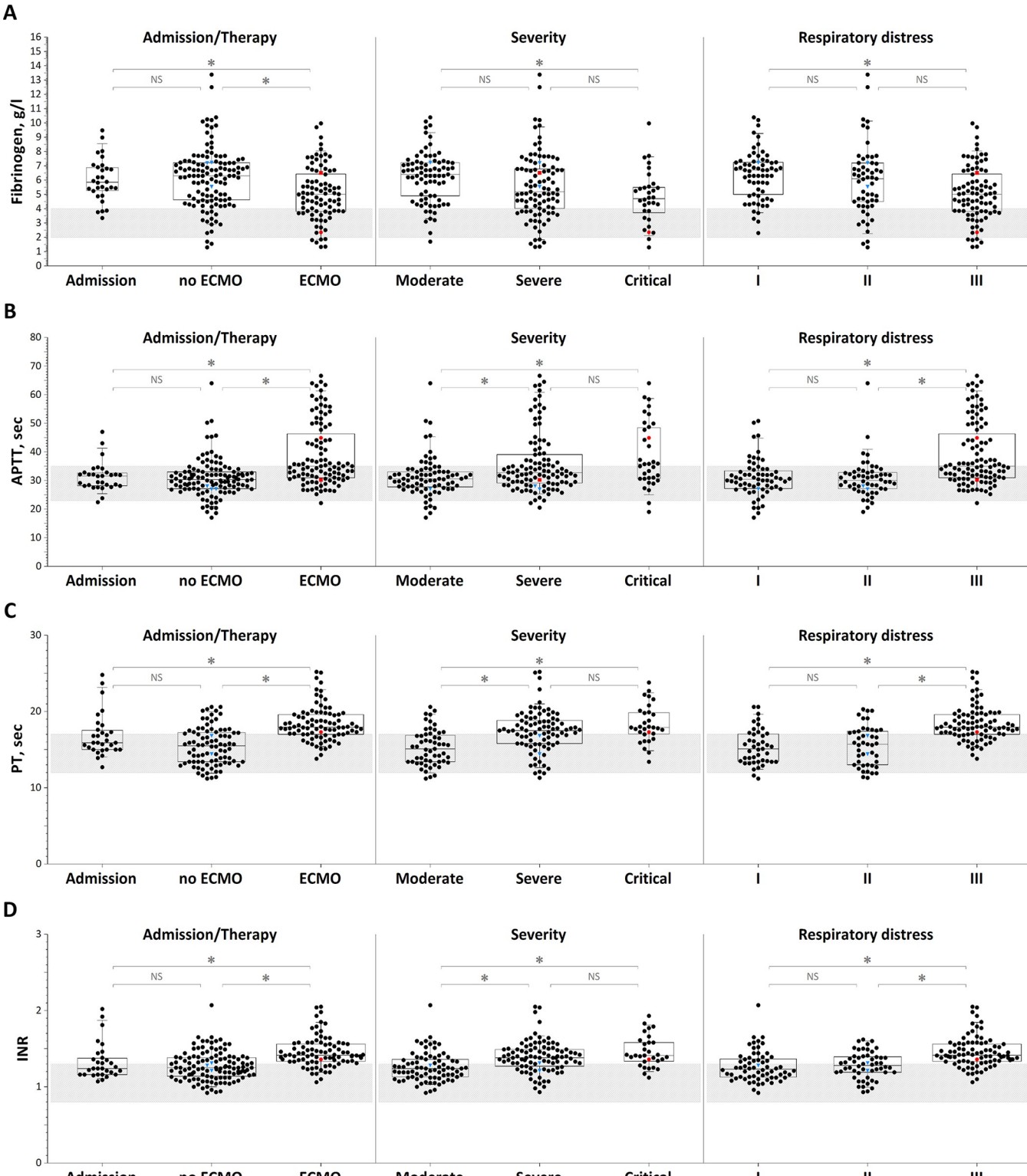

**Fig 3. Clotting assays.** (A) Fibrinogen, (B) APTT, (C) PT, and (D) INR in the patient groups: 1) on admission, with and without extracorporeal membrane oxygenation (ECMO) treatment; 2) depending on patient severity; and 3) depending on the severity of respiratory failure. The patterned area shows the normal reference ranges. The box plot indicates the following parameters: the median (the horizontal line inside the box), the 25th and 75th percentiles (the bottom and top of the box, respectively), and the 5th and 95th percentiles (the ends of the whiskers); * statistically significant difference (Mann–Whitney U-criterion, p<0.01). NS represents no significant difference. Red circle symbol–bleeding episodes followed in patients receiving heparin therapy, red square symbol–

bleeding episode followed in a patient with temporarily suspended heparin therapy, blue triangle symbol–thrombotic episodes followed in patients receiving heparin therapy.

episode (Vs = 10 μm/min, red square symbol) occurred in a patient who had a temporary suspension of heparin therapy, which also indicates that the test result was in the unsafe zone of hypocoagulation (the normal range for the speed of Vs is 20–29 μm/min for patients not receiving heparin therapy). Similarly, for episodes of thrombotic complications (blue triangle symbol), the clot growth rate (Vs) in all three cases was outside the range of effective and safe hypocoagulation (in two cases, despite the therapeutic doses of heparin, the patients were in the hypercoagulation zone). The results indicate a high prognostic value of the global thrombodynamics test in predicting thrombosis and bleeding.

Regarding TEG, in view of the absence of recommended target ranges for effective and safe hypocoagulation for TEG parameters, we correlated the results of measurements of the Angle parameter in TEG (as the parameter most sensitive to heparin) for all thrombohemorrhagic episodes with a range of normal values for this parameter. TEG was normal in 2 cases of bleeding and showed hypocoagulation in the third case (Fig 2B and 2C). For all cases of thrombosis, TEG was normal (Fig 2A and 2C).

## Discussion

The main conclusions of the study are as follows:

1. Integral assays, unlike classic assays, suggest severe hypercoagulation in up to 83% of COVID-19 patients on admission.

2. After admission, with LMWH treatment, there is a wide distribution of parameters suggesting the presence of both hypo- and hypercoagulation.

Our data agree with previous reports from Italian and American hospitals that patients with COVID-19 on admission have significant hypercoagulation by TEG and ROTEM [29, 37–39]. These whole blood-based rheological assays were the only type of integral or global assay that showed this, while classic clotting assays suggested hypocoagulation. In the present study, we used another assay of thrombodynamics, plasma-based and platelet-independent, which showed even higher hypercoagulation upon admission (in more than 80% of the

**Table 3. Correlation between TD and TEG parameters.**

| Spearman's $r_s$ | | Thrombodynamics | | | |
|---|---|---|---|---|---|
| | | **Vi, um/min** | **Vs, um/min** | **CS, um** | **D, a.u.** |
| **TEG: citrated native** | **R, min** | -0,66 [*] (n = 115) | -0,62 [*] (n = 115) | -0,67 [*] (n = 112) | 0,05 [NS] (n = 110) |
| | **K, min** | -0,65 [*] (n = 107) | -0,69 [*] (n = 107) | -0,73 [*] (n = 104) | 0,03 [NS] (n = 102) |
| | **Angle, deg.** | 0,70 [*] (n = 107) | 0,73 [*] (n = 107) | 0,78 [*] (n = 104) | 0,06 [NS] (n = 102) |
| | **MA, mm** | 0,48 [*] (n = 108) | 0,39 [*] (n = 108) | 0,50 [*] (n = 105) | 0,34 [*] (n = 103) |
| **TEG: citrated kaolin** | **R, min** | -0,61 [*] (n = 108) | -0,63 [*] (n = 104) | -0,63 [*] (n = 102) | -0,24 [NS] (n = 68) |
| | **K, min** | -0,62 [*] (n = 101) | -0,59 [*] (n = 97) | -0,64 [*] (n = 95) | -0,22 [NS] (n = 63) |
| | **Angle, deg.** | 0,62 [*] (n = 105) | 0,59 [*] (n = 101) | 0,63 [*] (n = 99) | 0,27 [NS] (n = 66) |
| | **MA, mm** | 0,22 [NS] (n = 104) | 0,24 [NS] (n = 100) | 0,25 [NS] (n = 98) | 0,51 [*] (n = 66) |

NS—not significant correlation

*—significant correlation (p = 0.01, Student's t-test)

n—number of points

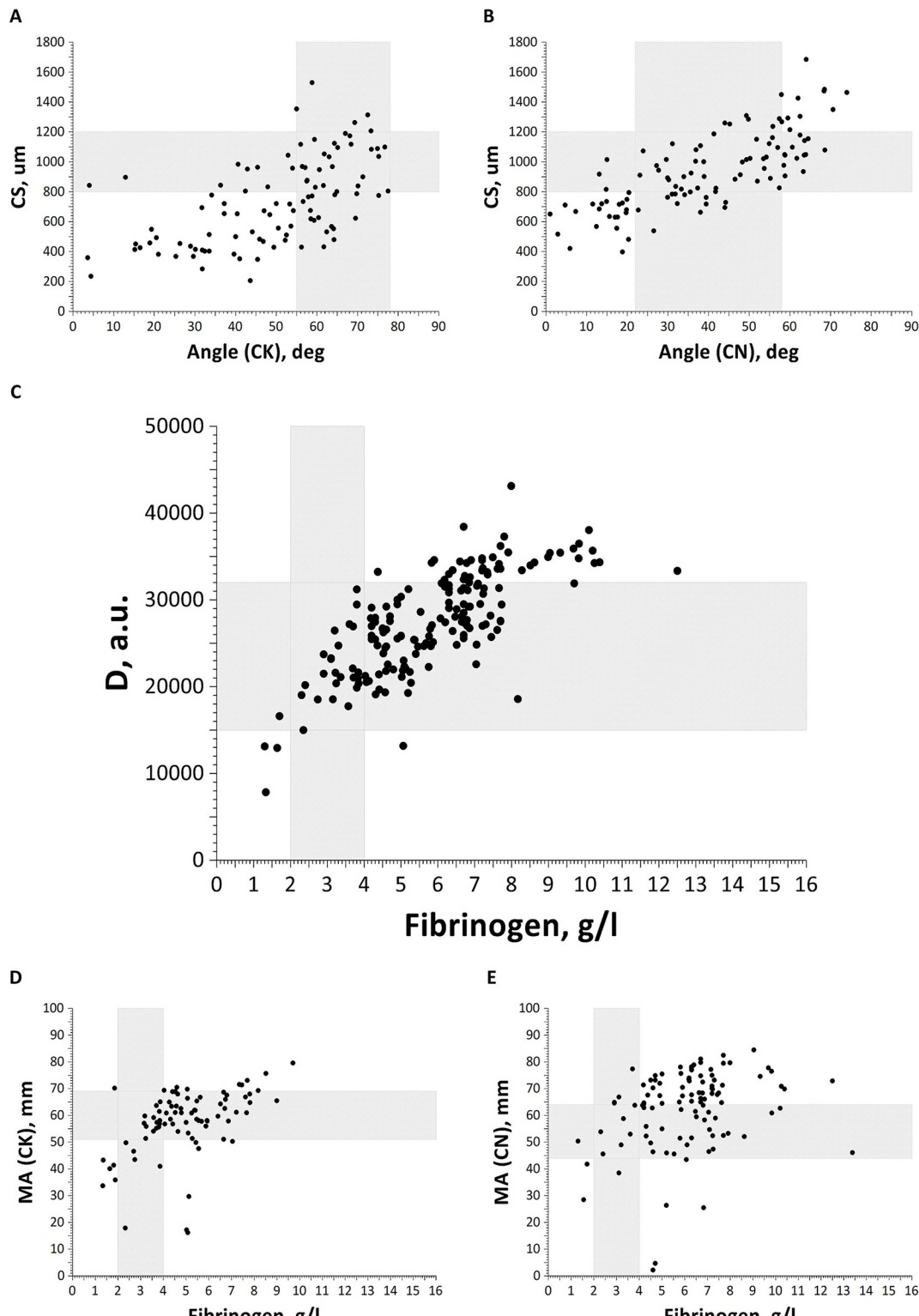

**Fig 4. Correlation plots.** (A) TD CS vs. TEG Angle (citrated kaolin), (B) TD CS vs. TEG Angle (citrated native), (C) TD D vs. fibrinogen, (D) TEG MA (citrated kaolin) vs. fibrinogen, and (E) TEG MA (citrated native) vs. fibrinogen. The patterned area shows the normal reference ranges.

**Table 4. Correlation of TD and TEG parameters with fibrinogen level.**

| Spearman's $r_s$ | Thrombodynamics | | | |
|---|---|---|---|---|
| | **Vi, um/min** | **Vs, um/min** | **CS, um** | **D, a.u.** |
| **Fibrinogen, g/l** | 0,10 [NS] (n = 208) | 0,07 [NS] (n = 205) | 0,11 [NS] (n = 200) | 0,74 [*] (n = 176) |
| | **TEG: citrated native** | | | |
| | **R, min** | **K, min** | **Angle, deg.** | **MA, mm** |
| | 0,08 [NS] (n = 113) | 0,04 [NS] (n = 105) | 0,07 [NS] (n = 105) | **0,33** [*] (n = 106) |
| | **TEG: citrated kaolin** | | | |
| | **R, min** | **K, min** | **Angle, deg.** | **MA, mm** |
| | 0,10 [NS] (n = 87) | -0,03 [NS] (n = 81) | 0,09 [NS] (n = 84) | 0,49 [*] (n = 83) |

NS—not significant correlation

*—significant correlation (p = 0.01, Student's t-test)

n—number of points

patients), while clotting assays remained in hypocoagulation in agreement with previous reports. Increased hypocoagulation was observed in ECMO patients and in those with more severe disease, likely due to additional heparin treatment.

In thrombodynamics, both clotting induced by the activator and activator-independent clotting were observed. Although the nature of hypercoagulation in COVID-19 is beyond the scope of the present study, it is important to note that thrombodynamics showed an increased rate of clot growth but not spontaneous clotting (unlike what has been reported for other disorders [36, 40]. This indicates an increased tendency of blood to clot rather than the presence of circulating activators, such as activating microparticles [40].

We hypothesize that the observed hypocoagulation in the more severely ill patients was more pronounced due to the increased heparin dosages. At the same time, the decrease in fibrinogen in severely ill patients was probably associated with liver failure. A significant limitation of this study is that patients receiving treatment were analyzed at a single point and with different LMWHs. However, their number was significant, and the doses used were consistently high. Both thrombodynamics and TEG data consistently showed the efficiency of therapy in suppressing hypercoagulation but also suggested the presence of a significant number of patients who had excessive or insufficient therapy. Although the data on clinical outcomes are in agreement with this view, additional studies are needed to ascertain statistical significance. The data indicate that integral assays can be an important tool for monitoring the status of patients and adjusting therapy. Prospective studies are needed to validate these suggestions.

## Supporting information

**S1 File.**
(PDF)

**S2 File.**
(DOCX)

**S3 File.**
(DOCX)

## Acknowledgments

We thank Tatyana Popova for collecting and maintaining the database and Alexey Ivanov for performing the statistical analysis and plotting the results.

## Author Contributions

**Conceptualization:** Andrey Y. Bulanov, Aleksandr G. Roumiantsev, Fazoil I. Ataullakhanov, Sergey S. Karamzin.

**Data curation:** Ekaterina L. Bulanova, Irina B. Simarova, Elizaveta A. Bovt, Olesya O. Eliseeva.

**Formal analysis:** Andrey Y. Bulanov, Sergey S. Karamzin.

**Investigation:** Irina B. Simarova, Elizaveta A. Bovt, Olesya O. Eliseeva.

**Project administration:** Aleksandr G. Roumiantsev, Fazoil I. Ataullakhanov.

**Writing – original draft:** Mikhail A. Panteleev.

**Writing – review & editing:** Elizaveta A. Bovt, Soslan S. Shakhidzhanov, Mikhail A. Panteleev, Sergey S. Karamzin.

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
