## [Decision Letter · Decision Letter 0]

9 Jun 2021

PONE-D-21-05154

Between heparin and hypercoagulation: integral assays of hemostasis in patients with COVID-19

PLOS ONE

Dear Dr. Bovt,

Thank you for submitting your manuscript to PLOS ONE. After careful consideration, we feel that it has merit but does not fully meet PLOS ONE’s publication criteria as it currently stands. Therefore, we invite you to submit a revised version of the manuscript that addresses the points raised during the review process.

We look forward to receiving your revised manuscript.

Kind regards,

Arijit Biswas

Academic Editor

PLOS ONE

Journal Requirements:

Additional Editor Comments:

2.In the ethics statement in the manuscript and in the online submission form, please provide additional information about the patient records/samples used in your study, including: a) whether data were collected prospectively for the purposes of research, or were collected routinely and accessed retrospectively; b) whether all data were fully anonymized before you accessed them; c) the date range (month and year) during which patients' medical records/samples were accessed; d) the date range (month and year) during which patients whose medical records/samples were selected for this study sought treatment; and e) the source of the medical records/samples analyzed in this work (e.g. hospital, institution or medical center name).

In addition, we note that you obtained consents from participants to take part in your study. In the Ethics Statement on the online submission form and the manuscript Methods , please clarify the context in which consent was obtained, and specify whether patients provided:

    1) Consent to use their medical records/samples in research

    2) Consent to undergo the procedure

    3) Consent to take part in the study reported in this manuscript.

Please also state the type of consent obtained (written or verbal). If the ethics committee waived the need for additional informed consent, please state this.

Thank you for your attention to these requests.

3. To comply with PLOS ONE submission guidelines, in your Methods section, please provide additional information regarding your statistical analyses. For more information on PLOS ONE's expectations for statistical reporting, please see https://journals.plos.org/plosone/s/submission-guidelines.#loc-statistical-reporting.

Additional Editor Comments:

It is clear from the reviewers comments that the article has publishing value but at the moment suffers from several flaws. This includes absence of important coagulation data, poorly defined coagulation parameters as well as flaws in the way the whole article has been presented. I would agree with all the reviewers and suggest that the authors revise their article in its entirety specially keeping in mind the points raised by the reviewers.

Reviewers' comments:

Reviewer's Responses to Questions

**Comments to the Author**

1. Is the manuscript technically sound, and do the data support the conclusions?

Reviewer #1: Yes

Reviewer #2: Partly

Reviewer #3: Partly

2. Has the statistical analysis been performed appropriately and rigorously? 

Reviewer #1: N/A

Reviewer #2: I Don't Know

Reviewer #3: I Don't Know

3. Have the authors made all data underlying the findings in their manuscript fully available?

Reviewer #1: Yes

Reviewer #2: Yes

Reviewer #3: Yes

4. Is the manuscript presented in an intelligible fashion and written in standard English?

Reviewer #1: No

Reviewer #2: No

Reviewer #3: No

5. Review Comments to the Author

Reviewer #1: The Authors performed an observational study on 153 hospitalized COVID-19 patients with different degrees of severity. Both routine clot assays and integral assays, such as TD and TEG, were performed in all the patients. Only measurements taken on the peaks of LMWH or UFH treatment were considered for the comparison analysis, while all the measurements of routine assays were taken in account. The data observed are in line with those collected by previous study which underlined a substantial different between routine assays which show an hypocoagulation state in COVID-19 patients while integral assays point out mostly an hypercoagulation state when patients were not under treatment. Moreover, after both LMWH or UFH treatment there are still some patients which show an hypercoagulation state or even an hypocoagulation (mostly due to UFH treatment, as the authors stated). Thus, the authors indicate a potential importance of integral assays to evaluate the blood coagulation imbalance in COVID-19 patients. However, while it’s very clear the clinical implication of TD assays in clinical management, apparently it’s not equally clear for TEG assays (i.e. there isn’t a target range of effective hypocoagulation nor prognostic value in predicting thrombosis and bleeding). Also, it may help if the Authors clarify their choice to use a plasma-based and platelet-independent thrombodynamics assay instead of whole-bloode based one as previous studies. Some revision of the English language is recommended.

Reviewer #2: Major issues:

1. It is unclear why the correlation of fibrinogen level with the admission/therapy, severity and respiratory disstress was studied but no other coagulation factor was analysed. The authors are encouraged to perform coagulation factor analysis such as thrombin, factor VII, factor X, as well as D-dimer and antithrombin III measurements and to monitor the correlation of the data with admission/therapy, severity and respiratory disstress.

Minor issues:

1. Some abbreviation such as ECMO and NEWS scale were not described properly at the first time mentioned in the text.

2. The resolution of the figure 1 (C-F), figure 2 (A-D) and fgure 3 (A-D) is not enough. The symboles used to clarify the results of statistical analysis are not readable.

3. In Table 1, the number of the patients and the age rows can be omitted as all of the data presented there are shown as n/a. In the same table the empty boxes should be filled properly, especially for TEG, citrated native, for severity critical group one measurement was done but the data is not shown.

4. In the table 2, no box is allowed to be left empty. The reason why the TEG using citrated kaolin was not perfomd on patients’ samples on admission should be described below the table.

5. In the table 3 and table 4 the same as figures the p values correspond to stars should be mentioned below the tables or in the figure legend.

6. The statistical analysis was used to analyse the results is not mentioned in the figure legengs.

Reviewer #3: Bulanove and his colleagues evaluated standard clotting assays as well as Thrombodynamics and Thromboelastography tests in COVID-19 patients, either on admission or on treatment. The data are not presented well and the manuscript is not written clearly. Then manuscript requires significant modifications/improvements, particularly, presenting results and discussing the data compared with the other studies in “Discussion” section.

Major comments:

- The introduction needs to be edited significantly. Until now there are several studies/data describing changes in coagulation parameters in COVID-19 patients (both ICU and non-ICU patients), including evaluating fibrinogen, D-dimer, VWF, PT, APTT, as well as reporting microthrombus in the postmortem biopsy samples, which should be addressed in this section of the introduction. The authors should give a review about changes in coagulation parameters in COVID-19 patients, both critical and non-critical patients, based on previous studies. In the second/last part of the introduction, the importance of the current study is not explained well.

- In Method/or Result section, the normal range for all the parameters should be defined clearly.

- In Method: what was the criteria for the using heparin treatment for the patients? Were they selected due to definitions for COVID-19 associated-coagulopathy? If yes what were these definitions (e.g. changes in PT, APTT, fibrinogen, TD, TEG…)?

- Results are not presented well, it is difficult to follow the finding, they are not well classified.

- Results, Page 5, lines 26-28 (results of figure 1) and page 6, lines 2-5 (results of the figure 2), explain why in severe cases there are hypocoagulation? Is it really a hypocoagulation (what about trend of changes in fibrinogen levels), or it is related to the TD and TEG assays used in this study?

- Discussion: The details of previous studies either using the same assays in COVID-19 patients, or impact of heparin treatment are not discussed sufficiently.

- Discussion: The reasons why the patients did not response to LMWHs therapy, according to the TD and TEG assays, are not discussed. In these patients, did the measuring the fibrinogen or D-dimer showed the same results like TD and TEG, after treatment?

Minor comments:

- The title of the manuscript is not appropriate, “Between heparin and hypercoagulation” is not meaningful, it should be changed.

- In Abstract, the aim of study is missing.

- In the abstract, the number of patients on treatment is missing.

- In the main text, whole manuscript, all abbreviations are not defined; at the first place, the abbreviations should be explained.

- In Method section, Line 8: please determine the “peak”, define how you decided for the “peak”.

- In legend of the images, all abbreviations should be defined, and the statistical analysis should be described, and the star significances in graphs should be defined, though the images was not clear and the stars/ns was not readable.

- The English writing (structure of the sentences) needs improvement: e.g. “Result” section, page 5, paragraph “Thrombodynamic assay in the patient groups”, the sentence starting with “The patients who….” is ambiguous, requiring changes.

6. PLOS authors have the option to publish the peer review history of their article (what does this mean?). If published, this will include your full peer review and any attached files.

Reviewer #1: No

Reviewer #2: No

Reviewer #3: No

---

## [Author Response · Author response to Decision Letter 0]

23 Jan 2023

Editor Comments:

Revised

2. In the ethics statement in the manuscript and in the online submission form, please provide additional information about the patient records/samples used in your study, including: a) whether data were collected prospectively for the purposes of research, or were collected routinely and accessed retrospectively; b) whether all data were fully anonymized before you accessed them; c) the date range (month and year) during which patients' medical records/samples were accessed; d) the date range (month and year) during which patients whose medical records/samples were selected for this study sought treatment; and e) the source of the medical records/samples analyzed in this work (e.g. hospital, institution or medical center name).

The patient’s data and blood samples were collected routinely in Moscow City Clinical Hospital №52 as a part of standard hospital treatment protocol. All data was anonymized in the hospital immediately when information was extracted from the patient's medical history. The patients whose medical records/samples were selected for this study sought treatment during April 2020 – June 2020. Patients' medical records/samples were accessed same time.

In addition, we note that you obtained consents from participants to take part in your study. In the Ethics Statement on the online submission form and the manuscript Methods , please clarify the context in which consent was obtained, and specify whether patients provided:

1) Consent to use their medical records/samples in research

2) Consent to undergo the procedure

3) Consent to take part in the study reported in this manuscript.

All the patients provided consent that their medical records/samples will be used in research. In case the blood samples were taken additionally (not in the scope of the routine hospital treatment protocol) the patients also provided consent to undergo the additional venipuncture procedure.

Please also state the type of consent obtained (written or verbal). If the ethics committee waived the need for additional informed consent, please state this. 

Regarding the consent of the patient. We usually obtained written consent from the patient. In the case of seriously ill patients verbal consent was obtained. 

3. To comply with PLOS ONE submission guidelines, in your Methods section, please provide additional information regarding your statistical analyses. For more information on PLOS ONE's expectations for statistical reporting, please see https://journals.plos.org/plosone/s/submission-guidelines.#loc-statistical-reporting.

Full dataset was used for statistical analysis, no outliers were removed. For groups comparison non-parametric Mann-Whitney U-criterion was used at the significance level of 0.01. For correlation analysis Spearman’s correlation coefficient was calculated at the significance level of 0.01. All the calculations were performed with the raw table data by self-written C# program and then double checked by means of SciPy – free and open-source Python library used for scientific computing and technical computing (https://www.scipy.org).

Revised

Review Comments to the Author:

Reviewer #1: The Authors performed an observational study on 153 hospitalized COVID-19 patients with different degrees of severity. Both routine clot assays and integral assays, such as TD and TEG, were performed in all the patients. Only measurements taken on the peaks of LMWH or UFH treatment were considered for the comparison analysis, while all the measurements of routine assays were taken in account. The data observed are in line with those collected by previous study which underlined a substantial different between routine assays which show an hypocoagulation state in COVID-19 patients while integral assays point out mostly an hypercoagulation state when patients were not under treatment. Moreover, after both LMWH or UFH treatment there are still some patients which show an hypercoagulation state or even an hypocoagulation (mostly due to UFH treatment, as the authors stated). Thus, the authors indicate a potential importance of integral assays to evaluate the blood coagulation imbalance in COVID-19 patients. However, while it’s very clear the clinical implication of TD assays in clinical management, apparently it’s not equally clear for TEG assays (i.e. there isn’t a target range of effective hypocoagulation nor prognostic value in predicting thrombosis and bleeding). Also, it may help if the Authors clarify their choice to use a plasma-based and platelet-independent thrombodynamics assay instead of whole-blood based one as previous studies. Some revision of the English language is recommended.

We thank the Reviewer for the assessment of the work and the important questions raised.

Truly, there is no target range of effective hypocoagulation nor prognostic value in predicting thrombosis for TEG. The reason why the TEG was used by doctors during the treatment is mostly not because of the need of estimation of the efficacy of heparin treatment, but because the need of estimation of the effect of transfusions of blood components or ECMO.

The main reasons why routine plasma-based TD assay was used instead of whole-blood based ones is its sensitivity for heparins and to hypo- and hypercoagulable states of plasmas.

Reviewer #2: Major issues:

1. It is unclear why the correlation of fibrinogen level with the admission/therapy, severity and respiratory disstress was studied but no other coagulation factor was analysed. The authors are encouraged to perform coagulation factor analysis such as thrombin, factor VII, factor X, as well as D-dimer and antithrombin III measurements and to monitor the correlation of the data with admission/therapy, severity and respiratory disstress.

There are several reasons why coagulation factor analysis such as thrombin, factor VII, factor X, as well as D-dimer and antithrombin III measurements were not performed:

1) We have studied only those coagulation tests that are approved for clinical use and are routinely available in city hospitals. That is why thrombin generation test was not used (not approved for clinical use in Russia). D-dimer, antithrombin III, anti-Xa and fVII assays usage is irregular in routine clinical practice because of its high cost. 

2) We have studied only those tests that may be used for monitoring the efficacy of heparin therapy effects. From previous data it is known that D-dimer, antithrombin III and fVII can’t be used for this purpose. Anti-Xa assay is very sensitive to heparin concentration, but not to its effect on hemostasis (depends on individual metabolic variations).

So the goal of the study was to compare two global assays that are only approved for clinical use and have the potential for estimation of COVID-19-induced hypercoagulation as well as heparin effects with the basic coagulation assay that are performed routinely for each and every patient in hospital.

Minor issues:

1. Some abbreviation such as ECMO and NEWS scale were not described properly at the first time mentioned in the text.

Revised 

2. The resolution of the figure 1 (C-F), figure 2 (A-D) and fgure 3 (A-D) is not enough. The symboles used to clarify the results of statistical analysis are not readable.

Revised 

3. In Table 1, the number of the patients and the age rows can be omitted as all of the data presented there are shown as n/a. In the same table the empty boxes should be filled properly, especially for TEG, citrated native, for severity critical group one measurement was done but the data is not shown.

Revised partially. Actually the number of the patients and the age rows can’t be omitted because these rows are filled with numbers at the first page of the table.

4. In the table 2, no box is allowed to be left empty. The reason why the TEG using citrated kaolin was not perfomd on patients’ samples on admission should be described below the table.

Revised 

5. In the table 3 and table 4 the same as figures the p values correspond to stars should be mentioned below the tables or in the figure legend.

Revised 

6. The statistical analysis was used to analyse the results is not mentioned in the figure legengs.

Revised 

Reviewer #3: Bulanove and his colleagues evaluated standard clotting assays as well as Thrombodynamics and Thromboelastography tests in COVID-19 patients, either on admission or on treatment. The data are not presented well and the manuscript is not written clearly. Then manuscript requires significant modifications/improvements, particularly, presenting results and discussing the data compared with the other studies in “Discussion” section.

Major comments:

- The introduction needs to be edited significantly. Until now there are several studies/data describing changes in coagulation parameters in COVID-19 patients (both ICU and non-ICU patients), including evaluating fibrinogen, D-dimer, VWF, PT, APTT, as well as reporting microthrombus in the postmortem biopsy samples, which should be addressed in this section of the introduction. The authors should give a review about changes in coagulation parameters in COVID-19 patients, both critical and non-critical patients, based on previous studies. In the second/last part of the introduction, the importance of the current study is not explained well.

Revised

- In Method/or Result section, the normal range for all the parameters should be defined clearly.

There are 16 different test’s parameters were analyzed giving 32 numbers for reference ranges. In order not to overfill main text with numbers and taking into account that there is no mentioning of absolute values of the test’s parameters in the text, all normal reference ranges for analyzed parameters were presented in Table 1.

- In Method: what was the criteria for the using heparin treatment for the patients? Were they selected due to definitions for COVID-19 associated-coagulopathy? If yes what were these definitions (e.g. changes in PT, APTT, fibrinogen, TD, TEG…)?

Yes, all patients received heparin because of COVID-19 diagnosis. Ministry of Health recommended heparin thromboprophylaxis for all hospitalized COVID-19 patients (the only contra indications – bleeding, low platelets count, florid renal failure). Routine hospital protocol was based on these recommendations.

- Results are not presented well, it is difficult to follow the finding, they are not well classified.

Revised

- Results, Page 5, lines 26-28 (results of figure 1) and page 6, lines 2-5 (results of the figure 2), explain why in severe cases there are hypocoagulation? Is it really a hypocoagulation (what about trend of changes in fibrinogen levels), or it is related to the TD and TEG assays used in this study?

Revised

We suppose that this is real hypocoagulation due to: 

1) Increased dosages of heparin used in these groups of patients (see table 1, «Daily dosage» column) 

2) Almost all severe cases were patients on ECMO (71%). Apart from higher heparin dosages used these patients suffer from low platelets count that additionally shifts coagulation to hypo-state. And in case of platelet-dependent tests, such as TEG, this low platelets level causes stronger hypocoagulation results. 

The Fibrinogen levels significantly differ only between «moderate» and «critical» groups. We do not know the exact reasons of lower fibrinogen levels in «critical» group and can only assume that this is because critical patients tends to suffer from liver dysfunction more often than moderately ill patients. The liver dysfunction causes decreased fibrinogen synthesis. 

- Discussion: The details of previous studies either using the same assays in COVID-19 patients, or impact of heparin treatment are not discussed sufficiently.

The study was conducted in April 2020 – June 2020 so there were no previous experience of using TD in COVID-19 patients. 

- Discussion: The reasons why the patients did not response to LMWHs therapy, according to the TD and TEG assays, are not discussed. In these patients, did the measuring the fibrinogen or D-dimer showed the same results like TD and TEG, after treatment?

Unfortunately, there was no possibility to track individual dynamics of coagulation parameters of patient after admission. The patients in group «Admission» and in group «Therapy» are different patients. But the data on figure 4 C-E shows that fibrinogen had correlated well with TD and moderately with TEG. D-dimer was not measured in this study.

Minor comments:

- The title of the manuscript is not appropriate, “Between heparin and hypercoagulation” is not meaningful, it should be changed.

Revised. 

New title: «Integral assays of hemostasis in hospitalized patients with COVID-19 on admission and during heparin thromboprophylaxis»

- In Abstract, the aim of study is missing.

Revised

Aim of the study: to investigate the status of the coagulation system by means of integral and local clotting assays in COVID-19 patients on admission to hospital and in hospitalized COVID-19 patients receiving heparin thromboprophylaxis.

- In the abstract, the number of patients on treatment is missing.

Revised

- In the main text, whole manuscript, all abbreviations are not defined; at the first place, the abbreviations should be explained.

Revised

- In Method section, Line 8: please determine the “peak”, define how you decided for the “peak”.

Revised. 

Median time of blood sampling after LMWH injection was 3.5 h, interquartile range - 0.5 h

- In legend of the images, all abbreviations should be defined, and the statistical analysis should be described, and the star significances in graphs should be defined, though the images was not clear and the stars/ns was not readable.

Revised

- The English writing (structure of the sentences) needs improvement: e.g. “Result” section, page 5, paragraph “Thrombodynamic assay in the patient groups”, the sentence starting with “The patients who….” is ambiguous, requiring changes.

Revised

---

## [Editor Report · Decision Letter 1]

28 Feb 2023

Integral assays of hemostasis in hospitalized patients with COVID-19 on admission and during heparin thromboprophylaxis

PONE-D-21-05154R1

Dear Dr. Bovt,

We’re pleased to inform you that your manuscript has been judged scientifically suitable for publication and will be formally accepted for publication once it meets all outstanding technical requirements.

Kind regards,

Arijit Biswas

Academic Editor

PLOS ONE
---

## [Editor Report · Acceptance letter]

24 May 2023

PONE-D-21-05154R1 

Integral assays of hemostasis in hospitalized patients with COVID-19 on admission and during heparin thromboprophylaxis 

Dear Dr. Bovt:

I'm pleased to inform you that your manuscript has been deemed suitable for publication in PLOS ONE. Congratulations! Your manuscript is now with our production department. 

Kind regards, 

on behalf of

Dr. Arijit Biswas 

Academic Editor

PLOS ONE